# Molecular and Physiological Determinants of Amyotrophic Lateral Sclerosis: What the DJ-1 Protein Teaches Us

**DOI:** 10.3390/ijms24087674

**Published:** 2023-04-21

**Authors:** Federica Sandrelli, Marco Bisaglia

**Affiliations:** 1Department of Biology, University of Padova, 35131 Padova, Italy; federica.sandrelli@unipd.it; 2Study Center for Neurodegeneration (CESNE), 35100 Padova, Italy

**Keywords:** amyotrophic lateral sclerosis (ALS), DJ-1, hypoxia, metabolism, mitochondria, oxidative stress

## Abstract

Amyotrophic lateral sclerosis (ALS) is an adult-onset disease which causes the progressive degeneration of cortical and spinal motoneurons, leading to death a few years after the first symptom onset. ALS is mainly a sporadic disorder, and its causative mechanisms are mostly unclear. About 5–10% of cases have a genetic inheritance, and the study of ALS-associated genes has been fundamental in defining the pathological pathways likely also involved in the sporadic forms of the disease. Mutations affecting the *DJ-1* gene appear to explain a subset of familial ALS forms. DJ-1 is involved in multiple molecular mechanisms, acting primarily as a protective agent against oxidative stress. Here, we focus on the involvement of DJ-1 in interconnected cellular functions related to mitochondrial homeostasis, reactive oxygen species (ROS) levels, energy metabolism, and hypoxia response, in both physiological and pathological conditions. We discuss the possibility that impairments in one of these pathways may affect the others, contributing to a pathological background in which additional environmental or genetic factors may act in favor of the onset and/or progression of ALS. These pathways may represent potential therapeutic targets to reduce the likelihood of developing ALS and/or slow disease progression.

## 1. Introduction

Amyotrophic lateral sclerosis (ALS) is an adult-onset disease characterized by the progressive degeneration of motoneurons (MNs) in the brain and spinal cord, which leads to the progressive paralysis of skeletal muscles and, ultimately, to death due to respiratory failure, within 3 to 5 years after the onset of disease symptoms [1]. One of the pathological hallmarks of ALS is the presence of cytoplasmic inclusions mainly composed of the protein TAR Binding-Protein 43 (TDP43), which have been found in the brain and spinal cord neurons of most ALS patients [1]. Despite decades of research, causative pathogenic mechanisms in ALS are still partially elusive. It is likely that both genetic susceptibilities and environmental factors participate in the onset and progression of the disease [1]. Although ALS is mainly a sporadic disorder, approximately 5–10% of ALS cases have a genetic inheritance, and the analysis of the genes involved in the disorder has contributed to the definition of potential pathological pathways that might also participate in the sporadic forms of the disease.

The first gene that has been linked to familial forms of ALS is the gene coding protein superoxide dismutase 1 (SOD1), a fundamental Cu/Zn metalloenzyme involved in the control of cellular redox homeostasis [2]. To date, about 200 *SOD1* gene modifications have been identified [3,4]. Pathogenic mutations are distributed throughout the entire sequence, and all of them accentuate the structural instability of the apo form of the protein, promoting the formation of disordered SOD1 conformers and leading to the accumulation of intracellular aggregates [3,4]. Interestingly, data obtained in ALS mouse models, based on the pathological G37R or G93A SOD1 mutations, highlighted copper incorporation into the SOD1 active site, rather than amino-acid mutations, per se, as a major determinant for motor neuron degeneration [5,6,7]. Importantly, the role of dysfunctional apo-SOD1 in ALS pathology has been further supported in a recent study through detailed examinations of disease-affected tissues from ALS patients [8].

Mutations in the gene encoding the TDP43 protein have been also linked to familial forms of ALS [9,10,11], further highlighting the role of the protein in the pathogenesis of ALS. TDP43 is a DNA/RNA-binding protein, which plays numerous roles in RNA metabolism [1]. Under physiological conditions, it is mainly a nuclear protein, while, in a pathological state, TDP43 shows nuclear depletion with concomitant cytosolic accumulation, protein truncation, and hyperphosphorylation with subsequent aggregation, so that both the loss of normal functions in the nucleus and the toxic gain of functions in the cytoplasm are considered to participate in TDP43 proteinopathy [1,12].

In 2005, three Italian brothers with a *DJ-1* gene mutation were reported to be affected by parkinsonism, dementia, and ALS-like symptoms, including muscle atrophy of the upper and lower limbs with diffuse fasciculation and denervation. [13]. As extensively reported below, DJ-1 is a ubiquitous protein which has been implicated in multiple cellular functions, including antioxidative responses and transcription [14]. The direct sequencing of the *DJ-1* gene displayed a missense mutation that resulted in an amino acid substitution of lysine with glutamic acid (E163K) [13]. More recently, similar pathological phenotypes with diffuse denervation and fasciculations in the four limbs, and lower motor neuron impairment, were reported in a consanguineous Turkish family carrying a new *DJ-1* mutation, which causes a stop codon in position 45, with the consequence of no protein expression [15]. Interestingly, DJ-1 levels in cerebrospinal fluid collected from sporadic ALS patients were found to be significantly higher than in control subjects [16], and, consistently, DJ-1 immunostaining was reported to be stronger in the human motor cortex and spinal cord sections of ALS patients than in controls [17]. While this observation could appear counterintuitive considering that the loss of DJ-1 function is associated with familial forms of ALS, it is possible that DJ-1 upregulation represents a first compensatory response to cope with the abnormalities occurring during the initial stages of the disease.

When considering experimental models of the disease, in a mouse model based on the overexpression of the human G93A SOD1 protein, *DJ-1* knock-out was shown to lead to an accelerated disease course and shortened survival time [18]. In contrast, DJ-1 over-expression resulted in increased cell viability and reduced cell toxicity in G93A SOD1-transfected neuronal cells [16] and was demonstrated to protect against cellular damage induced by the aberrant aggregation of TDP-43 [19].

A strong indication of the possible involvement of DJ-1 in ALS comes from the recent approval in the US and Canada of a new drug for the treatment of ALS, composed of a fixed-dose combination of taurursodiol and sodium phenylbutyrate. Taurursodiol has been described to recover mitochondrial bioenergetic deficits by inhibiting the mitochondrial permeability and increasing the apoptotic threshold of the cell [20]. Sodium phenylbutyrate, instead, is a histone deacetylase inhibitor that has been indicated to enhance the expression level of heat shock proteins, thereby reducing the toxicity that could arise from endoplasmic reticulum stress [20]. Interestingly, sodium phenylbutyrate has been described to mediate neuronal protection in both cellular and animal models of Parkinson’s disease and ischemia/reperfusion injury through the upregulation of the DJ-1 protein [21,22].

DJ-1 is highly expressed in the central nervous system (CNS), and, in particular, at the astrocytic level [23]. The possible involvement of astrocytes in ALS pathogenesis has been extensively discussed elsewhere [24] and is out of the scope of this review. However, it is worth mentioning that the high presence of DJ-1 at the astrocyte level under physiological conditions could imply that DJ-1 misregulation in this brain cellular population might be somehow related to the disorder. Several functions have been ascribed to DJ-1, including the modulation of signaling pathways through the regulation of several transcription factors, the modulation of glucose levels, chaperone activity responsible for protein folding, and glutathione-independent glyoxalase activity, as well as deglycase activity, both involved in carbonyl stress defense [14,25,26,27]. Nonetheless, the most corroborated and accepted physiological role of DJ-1 is its participation in the cellular antioxidant response. In this frame, as described below, DJ-1 participates in mitochondria homeostasis and in the preservation of reactive oxygen species (ROS) physiological levels [28]. While DJ-1 functions seem to be redundant, they could acquire relevance under stressful conditions, and the DJ-1 multimodal response to different sources of stress might represent the way DJ-1 exerts its neuronal protection role. For these reasons and considering that the protein has been associated with very rare familial forms of ALS, the involvement of DJ-1 in ALS as a primary triggering factor can be considered of low clinical impact. Nevertheless, *DJ-1* mutations could represent a genetic background that makes individuals more sensitive to other fundamental pathological factors, and consequently more prone to ALS disease development. Thus, a better understanding of the cellular pathways involving DJ-1 could improve the knowledge of the mechanisms underlying the pathogenesis of this multifactorial disorder which may favor the onset of the disease. In this review, we focus on some of the physiological functions of DJ-1 that, when altered, could contribute to the onset and/or progression of ALS.

## 2. DJ-1 Protection against Oxidative Stress and Mitochondrial Dysfunction: Implications for ALS

As aforementioned, the most widely accepted function of DJ-1 is its protective role against excessive ROS levels. Accordingly, the overexpression of DJ-1 results in neuronal cytoprotection against oxidative damage, whereas DJ-1 deficiency leads to an increase in oxidative-stress-induced cell death, both in cell cultures and animal models [28,29,30,31,32,33,34,35,36]. In this regard, it has been suggested that DJ-1 may sense the cellular redox state through a highly conserved cysteine residue localized at position 106, contributing to the regulation of redox homeostasis [37].

In antioxidant defense, DJ-1 has been indicated to induce a large variety of responses to promote cell survival (Figure 1). For instance, DJ-1 has been described to participate in modulating several oxidative responsive signaling pathways by modulating gene expression through interaction with transcription factors and their regulators [14,38]. Accordingly, DJ-1 seems to activate the Extracellular-Signal-regulated Kinase 1/2 (ERK1/2) pathway, which stimulates cell survival in the presence of several stimuli, such as Tumor Necrosis Factor (TNF), growth factor withdrawal, and nitric oxide [14]. From an ALS perspective, it is worth mentioning that, by promoting ERK1/2 nuclear translocation and the following activation of the transcription factor ETS-domain-containing protein 1 (Elk1), DJ-1 has been described to increase SOD1 transcription under oxidative conditions [39] (Figure 1). DJ-1 has also been reported to modulate the activity of the Nuclear factor E2-Related Factor 2 (Nrf2) transcription factor by promoting the dissociation of Nrf2 from its inhibitor Kelch-like ECH-Associated Protein 1 (Keap1), thus inducing Nrf2 nuclear translocation and the expression of its target genes [40,41,42]. Nrf2 is a master regulator of multiple cytoprotective responses. It participates in the transcription of numerous genes involved in glutathione metabolism, iron homeostasis, and antioxidant response as well as in the regulation of energetic metabolism, including glycolysis, pentose phosphate pathway, fatty acid, and glutamine metabolism. Interestingly, Nrf2 expression has been described to decrease with aging, and, more importantly, the Nrf2 pathway appears to be compromised in ALS patients as well as in cell cultures and animal models of the disease [43,44,45,46].

Always related to ALS, DJ-1 has been shown, in vitro, to bind copper [47,48,49] and to interact with and activate SOD1 through copper transfer [49,50], suggesting the possibility that the antioxidant activity of DJ-1 is mediated by SOD1. This hypothesis, however, needs further investigation since a recent study performed in *Drosophila melanogaster* was unable to confirm the participation of DJ-1 in the SOD1 maturation pathway in vivo [29], while the interaction between copper and DJ-1 was not detected in an NMR analysis performed in a cellular environment [51].

Besides the regulation of signaling cascades, several studies have demonstrated the role of DJ-1 in the maintenance of mitochondrial homeostasis (Figure 1). Accordingly, DJ-1 deficiency is associated with several mitochondrial morphological and functional phenotypes, including mitochondrial fragmentation, reduced calcium uptake, impaired respiration, and altered membrane potential [28,52,53,54,55,56,57]. Moreover, DJ-1 has been described to participate in the process of mitochondrial quality control, being involved in the Parkin/PINK1-mediated mitophagy [54,58,59,60], although conflicting results exist in the literature concerning the molecular mechanisms of such involvement. Interestingly, convincing evidence indicates that under oxidative stress, DJ-1 translocates into mitochondria, thus exerting its antioxidant and neuroprotective action in this organelle [61,62,63].

Collectively, these results show that DJ-1 participates in the regulation of the cellular redox state and in the maintenance of mitochondrial homeostasis, supporting the already existing experimental evidence that the interplay between mitochondrial dysfunctions and oxidative stress could contribute to favoring the onset and/or the progression of the disease. In fact, as recently reviewed, mitochondrial alterations, including calcium dyshomeostasis, decreased mitochondrial bioenergetic capacity, and defects in mitochondrial respiratory chain complexes have been observed in both the motoneurons and spinal cords of ALS patients [64,65] (Figure 1). Considering the post-mitotic nature of these neuronal cells, they might be very sensitive to the oxidative injury that accumulates during their lifespan.

Consistent with the detrimental role of redox imbalance in ALS pathology, edaravone, one of the very few drugs available to treat the disease, has been described as an ROS scavenger [64]. More specifically, although the precise mechanism of action of edaravone is still partially unknown, the compound has been shown to trap hydroxyl radicals and peroxynitrite anions and decrease the accumulation of hydrogen peroxide through the upregulation of Peroxiredoxin-2 [64]. In this frame, considering the purported role of DJ-1 in activating the Nrf-2 antioxidative pathway and the possible involvement of this pathway in ALS, the development of effective Nrf-2 activators appears a promising therapeutic strategy to cope with ALS, which deserves consideration [43,44,45,46].

## 3. DJ-1 Involvement in Energy Metabolism: Implications for ALS

One of the main functions of mitochondria is the synthesis of ATP to satisfy the cellular energetic demand. This process is essential for the effective functioning of high-energy-demanding organs, such as the brain, heart, and skeletal muscles. In particular, with MNs being characterized by long axons, they require high mitochondrial ATP production to transport cellular components from the soma to the axonal terminal, maintain the membrane potential throughout the axon, and sustain the neurotransmission process. For these reasons, MNs are particularly vulnerable to mitochondrial dysfunction. Considering the participation of DJ-1 in mitochondrial homeostasis, the potential involvement of the protein in the regulation of cellular metabolism could have relevant pathological implications.

Although a unifying picture is still lacking and additional analyses are necessary to resolve controversial aspects, all the information currently available highlights the participation of DJ-1 in the modulation of several aspects directly related to glucose homeostasis. At least three independent studies observed an association between DJ-1 deficiency and decreased ATP levels, either in cell or *D. melanogaster* models [53,59,66], probably due to mitochondrial complex I impairment [53]. In contrast with these studies, despite a decreased complex-I-linked oxygen consumption rate, we were unable to detect any variation in ATP and/or ADP levels in *DJ-1* knock-out flies under either fed or starved conditions [28]. However, very interestingly, we observed that DJ-1-deficient flies exposed to starvation showed premature mortality, suggesting that while under fed conditions, the activation of some compensatory mechanisms, such as glycolysis, gluconeogenesis, and/or fatty acid oxidation, may contribute to the maintenance of ATP levels, they fail under prolonged starvation [28]. In other words, in the presence of a mild mitochondrial impairment induced by the absence of DJ-1, alternative energetic reservoirs are utilized, involving glycogen degradation, protein breakdown, or lipolysis, while in the presence of additional stress factors, the compensation fails. Consistent with this hypothesis, a proteomic analysis carried out on DJ-1-deficient mice revealed significant alterations in the levels of several proteins involved in energy-production-related pathways, such as the glycolysis and tricarboxylic acid cycle (TCA), in comparison to control mice [59]. In particular, decreased levels of pyruvate dehydrogenase, a key enzyme that connects glycolysis to the TCA, were highlighted by the investigators. Interestingly, through a series of unbiased high-content approaches to measuring brain biomolecules in two rodent models, one of the first studies focusing on the effects of DJ-1 on metabolism showed that a lack of DJ-1 causes a redistribution of hexokinase 1 from mitochondria to the cytosol, accompanied by a shift in glucose metabolism from glycolysis to the polyol pathway. [67]. The involvement of DJ-1 in the glycolytic pathway also emerged from other studies in which the loss of DJ-1 function has been described to significantly increase the activities of key regulatory glycolytic enzymes, suggesting the hypothesis that glycolysis may be enhanced by DJ-1 deficiency [66,68]. Besides its influence on glycolysis, the loss of DJ-1 in mouse models has also been reported to reduce adipogenesis and body weight, together with low glucose levels and insulin resistance [69,70,71], suggesting that lipolysis may be affected by the protein, as well. Always related to the impact of DJ-1 on metabolism, a recent metabolomic analysis performed in DJ-1-deficient flies showed alterations in protein metabolism. More specifically, among the amino acids with levels that were reduced in flies lacking DJ-1, the investigators found branched-chain amino acids (BCAAs) [72]. Since BCAAs play several important metabolic and physiological roles and are substrates for the synthesis of proteins [73], these results appear especially relevant in the frame of ALS, where the impairment in muscular protein synthesis might exacerbate the pathological symptoms.

Although a comprehensive model is far from being achieved, the experimental evidence accumulated so far supports the participation of DJ-1 in the control of cellular metabolism. The loss of DJ-1 function seems to confer subtle bioenergetic impairments, which can be exacerbated and become more relevant under conditions of stress. These results support the notion that dysregulation in energy homeostasis might contribute to ALS onset and progression. Interestingly, as extensively discussed in a recent review, premorbid weight loss has been often observed in ALS patients as an indication of defective energy metabolism, and the risk of developing the disease is higher in people with lower premorbid body fat [74]. On the basis of these observations, it has been suggested that an abnormal utilization of nutrients might trigger weight loss [74]. Consistent with this hypothesis, in ALS patient-derived myotubes, alterations in glucose and fatty acid oxidation were linked to the resting energy expenditure of ALS patients and disease progression [75]. Moreover, diminished glucose metabolism has been repeatedly reported in several studies, carried out using fibroblasts derived from ALS patients, in which the investigators observed reductions in enzymes or substrates involved in glycolysis and a compensatory increase in fatty acid oxidation to generate ATP via β-oxidation [76,77]. Similar results were also obtained through whole-genome expression profiling performed in the post-mortem cortex of sporadic ALS patients [78].

Interestingly, a high-energy diet gave protection in two mouse models carrying either G86R SOD1 or A315T TDP43 pathological mutations [79,80], and some encouraging results were also obtained in two clinical trials. In the first study, based on 24 participants, patients who received a high-carbohydrate hypercaloric diet were less likely to have serious adverse events, including death, during the 5-month follow-up with respect to an isocaloric diet [81]. While the results of the second study provided no evidence of a life-prolonging effect of a high-caloric fatty diet for the whole ALS population, a significant survival benefit for the subgroup of fast-progressing patients was observed [82]. In light of these promising results, further studies are warranted to explore the possible protective effects of a well-calibrated diet on the progression of the disease. Moreover, BCAA supplementation might also be investigated as a potential therapeutic option. In this regard, ketone bodies or ketogenic diets have shown protective effects in mutant SOD1 mouse [83] and fly models of TDP-43 ALS [84] and remain to be tested further.

## 4. DJ-1 Involvement in Hypoxia Response: Implications for ALS

Recently, using *D. melanogaster* as an in vivo model, we have demonstrated that the absence of DJ-1 affects adult fly survival under oxygen depletion [29], in line with a purported protective role of DJ-1 against hypoxia [27]. One of the key events occurring in response to hypoxia is the activation of the transcription factor Hypoxia-Inducible Factor-1α (HIF-1α), which otherwise, under normal oxygen tension, is constantly hydroxylated by the enzyme prolyl hydroxylase (PHD) to be recognized and labeled by the E3 ubiquitin ligase Von Hippel–Lindau (VHL) and then degraded at the proteasome [27] (Figure 2). Under limited oxygen availability, HIF-1α is stabilized and interacts with the HIF-1β subunit to form a constitutively active complex that translocates into the nucleus where it binds to specific hypoxia-responsive elements (HREs), promoting the expression of numerous genes involved in hypoxic adaptation, including those participating in vascularization and the reprogramming of energy metabolism [27]. In fact, while under normoxia, neurons largely rely on oxidative phosphorylation for ATP synthesis, and hypoxic ATP production mainly depends on glycolysis. Accordingly, hypoxia-induced HIF-1α activation leads to the upregulation of glycolytic enzymes and glucose transporters to promote glycolysis [85], with a concomitant attenuation of the electron flux through the mitochondrial respiratory chain, which has been demonstrated to be at least partially due to the inhibition of complex I activity [86].

Although a general agreement has not been reached at the mechanistic level and contrasting results have been reported [27], numerous pieces of evidence indicate a protective role of DJ-1 against ischemic injury. Indeed, in vivo analyses have indicated that the absence or downregulation of DJ-1 increased the sensitivity to ischemia and enhanced the infarct size in the brain [22,87,88,89,90] and heart [91,92]. In contrast, DJ-1 upregulation has been described to rescue post-ischemic cerebral [87,89,90,93,94] and myocardial damage [95,96,97]. The principal protective mechanism promoted by DJ-1 against ischemia seems to be related to the stabilization of HIF-1α (Figure 2). Actually, reduced levels of HIF-1α have been observed under hypoxia following DJ-1 silencing in osteosarcoma-derived cells [98]. DJ-1 has been suggested to interact with VHL by acting as a negative regulator of the ligase, protecting HIF-1α from proteasomal degradation in such a way [99]. Alternatively, DJ-1 has been reported to induce HIF-1α accumulation in colorectal cancer cells through the activation of the PI3K/Akt transcriptional pathway [100], suggesting the possibility that different mechanisms could be activated in different tissues.

Interestingly, alterations in the expression profile of specific genes controlled by HIF have been described in ALS patients and in animal models of the disease. While in most cases, such alterations are secondary events since they are the result of hypoxic conditions arising from dysfunctions in the respiratory muscles, some indications suggest that changes in hypoxia signaling and adaptation might contribute to ALS onset and/or progression [64]. In particular, lower levels of vascular endothelial growth factor (VEGF) were measured in the cerebrospinal fluid (CSF) of early ALS patients (i.e., during the first year of the disease), compared with controls [101]. VEGF expression, which is under the control of HIF, stimulates vessel growth and promotes the survival of motor neurons during hypoxia (Figure 2) [101]. The same investigators also observed a paradoxical response, with a lack of VEGF upregulation in the CSF of hypoxemic ALS patients in comparison to control individuals, represented by hypoxemic patients with other neurological disorders [102,103]. Similar results were subsequently obtained by evaluating VEGF production in monocytes derived from ALS patients during either acute or prolonged hypoxia. Interestingly, low VEGF levels were measured in the early stages of the disease (at the time of the diagnosis), before the appearance of any severe respiratory muscle atrophy and prolonged hypoxemia [104]. Concomitant with altered VEGF levels, the overactivation of PHD-2 was also observed, suggesting that a partial lack of protection mechanisms associated with hypoxia might be present in ALS patients [104].

The involvement of DJ-1 in the response to hypoxia appears extremely relevant considering the functions ascribed to DJ-1 in the control of mitochondrial and redox homeostasis, as well as in energy metabolism. As previously discussed, mitochondrial dysfunction can dramatically increase the production of ROS while decreasing the energy supply to motoneurons, which may aggravate the progression of ALS. Since hypoxia activates an adaptive response to allow cells to survive in the presence of limiting oxygen levels and decreases the reliance of cells on mitochondrial oxidative metabolism, this metabolic response may be exploited to counter ALS progression. More specifically, the data presented here indicate that the modulation of HIF activity might represent a promising novel therapeutic approach against ALS progression. HIF stabilization, for instance, may protect motoneurons by inducing the expression of neuroprotective and neurotrophic genes or by enhancing glycolysis through the activation of glycolytic enzymes and glucose transporters. Interestingly, in a genetic mouse model of Leigh syndrome, which represents the most common form of pediatric mitochondrial disease, chronic hypoxia led to a marked improvement in pathological phenotypes, indicating that the hypoxic response may induce protective effects on mitochondrial toxicity [105]. Furthermore, chronic hypoxia or intermittent hypoxic conditioning was beneficial in several models of neurodegenerative or mitochondrial-related disorders, such as Leigh syndrome, Friedreich’s ataxia, and mild cognitive impairment [85]. However, further studies are necessary to evaluate whether hypoxic exposure can represent an effective protective treatment for ALS. In fact, as recently reviewed, the late stages of the disease seem to be characterized by chronic HIF stabilization that, in turn, appears to promote increased levels of ROS and pro-apoptotic factors, aggravating the symptoms of the disease [64].

## 5. Conclusions

Since the discovery that mutations in the gene coding the protein DJ-1 are associated with familial forms of PD, numerous studies were performed to define the physiological function of the protein. Nonetheless, despite large scientific efforts, a precise picture of what DJ-1 actually does is still missing. Many functions have been ascribed to the protein with the most accepted one being related to its ability to protect against redox alterations, although the molecular mechanisms underlying such a function are still elusive. On the basis of the published literature, it is plausible that DJ-1 possesses redundant protective functions that could become relevant in the presence of elevated stress conditions where DJ-1 could take part in the maintenance of cellular homeostasis. Consistent with this hypothesis, in the framework of ALS pathology, high DJ-1 levels have been detected in the cerebrospinal fluid of ALS patients, suggesting that its upregulation might represent a protective compensatory response. DJ-1 loss of function per se does not appear to be involved in the ALS onset as a primary event, and this is most probably the reason why its upregulation fails to stop the progression of the disease. However, DJ-1 misregulation could make individuals more prone to developing the disease when other risk factors are present as well. For this reason, a better understanding of the physiological function(s) of DJ-1 might shed some light on the molecular and cellular pathways that, when altered, might promote ALS onset.

Among the different roles ascribed to DJ-1, in this review, we have focused on some interconnected functions whose alterations can affect each other. More specifically, DJ-1 loss of function might promote mitochondrial dysfunction as well as ROS accumulation, two conditions that are strongly interconnected in a vicious cycle. Additionally, mitochondrial impairment might result in altered ATP levels, especially in highly demanding tissues, such as the brain and striatal muscles, and/or under sustained physical activity. Since the ability of an individual to cope with an increased oxygen demand or insufficient oxygen levels relies in part on the ability to activate a hypoxic response, considering the protective role that DJ-1 seems to exert under hypoxic conditions, the loss of its functionality might exacerbate defects in energy metabolism in addition to increasing ROS levels. Consequently, these cellular pathways may represent potential therapeutic targets to slow down the progression of the disease. Accordingly, as reported above, redox impairments represent the target of edaravone, one of the very few drugs approved to treat ALS. Considering the role of mitochondria as the main source of intracellular ROS, it is possible that mitochondria-directed antioxidants could be more effective than edaravone. In this frame, the mitoQ compound, first described in 2001 [106], showed neuroprotective effects in a mouse model of inherited ALS [107] and is currently the object of a clinical trial on multiple sclerosis patients (NCT04267926). Alternatively, molecules able to stimulate the Nrf2-mediated antioxidant response might be exploited as a therapeutic strategy to counteract oxidative stress in ALS patients. In this regard, metformin could represent a potential candidate. The molecule has been described to activate the Nrf-2 pathway and promote mitochondria functionality, and, very importantly, it is already in use to treat type-2 diabetes mellitus with minimal side effects [108]. Another pathway that could be the object of further investigation concerns energy metabolism. Given the encouraging results obtained in the presence of a high-carbohydrate hypercaloric diet, it would be interesting to examine the response to BCAA supplementation, which represents a still unexplored field.

## Figures and Tables

**Figure 1 ijms-24-07674-f001:**
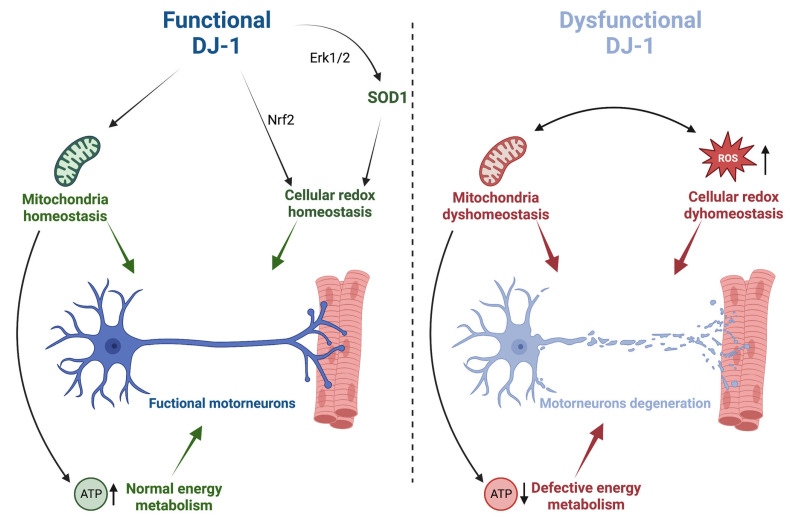
The physiological functions of DJ-1 possibly affected in ALS pathology. DJ-1 participates in the maintenance of mitochondrial and cellular redox homeostasis. These functions are fundamental to sustaining the high energetic demand of MNs without producing excessive levels of ROS, and collectively, they contribute to MNs’ healthiness (**on the left**). Alterations in DJ-1 physiological functions might affect mitochondria functionality, reducing the levels of ATP below MNs’ needs. At the same time, mitochondria dysfunctions might promote increased levels of ROS. Altogether, these alterations might constitute a pathological background enhancing the probability of developing ALS (**on the right**) (created with BioRender.com, accessed on 6 April 2023).

**Figure 2 ijms-24-07674-f002:**
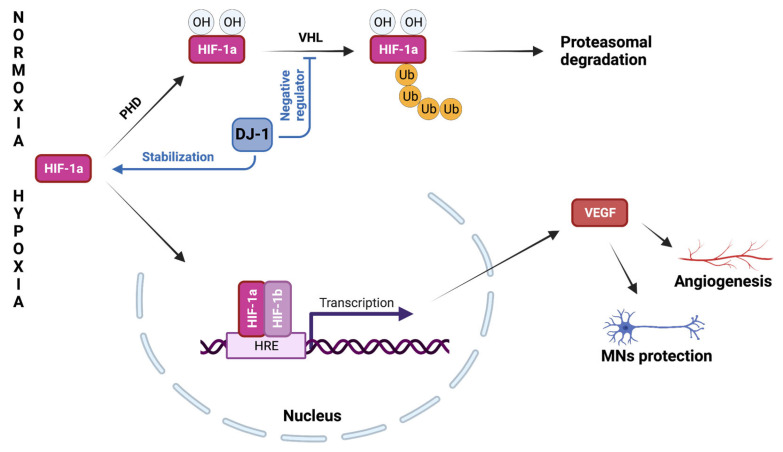
DJ-1-mediated protection against hypoxia. Under normoxia conditions, HIF-1α is constantly hydroxylated, ubiquitinated, and degraded at the proteasome. Under hypoxic conditions, HIF-1α translocates into the nucleus, where it promotes the expression of numerous genes involved in hypoxic adaptation. Among them, the VEGF protein stimulates vessel growth and promotes the survival of motor neurons during hypoxia. DJ-1 has been proposed to act as a negative regulator of VHL stabilizing the transcription factor HIF-1α. Acronyms are mentioned in the main text. (Created with BioRender.com, accessed on 6 April 2023.)

## Data Availability

Not applicable.

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
