# Peer review of "Molecular and Physiological Determinants of Amyotrophic Lateral Sclerosis: What the DJ-1 Protein Teaches Us"

_ijms, 2023, doi:10.3390/ijms24087674_

Round 1
Reviewer 1 Report
The authors have reviewed the biological functions of the protein DJ-1. The value of this review comes because mutations in the DJ-1 gene are linked with a subset of familial ALS forms. Understanding DJ-1 protein function may therefore help us understand how disrupting those functions contributes to ALS and, more broadly, may offer insight into which disease mechanisms drive all forms of ALS.
The review contains a great deal of valuable information which does indeed offer insights into potential disease mechanisms of ALS. The strongest sections of the review are 2, 3, and 4, which set out evidence supporting DJ-1 involvement in oxidative stress and mitochondrial function, energy metabolism, and response to hypoxia. The greatest room for improvement is in sections 1 and 5. The authors could make these sections more valuable and easier to read by being more concise and precise in setting out why understanding DJ-1 mutation (over other genetic associations) is an especially rich source of insight into general ALS disease mechanisms (section 1); and setting out more clearly which insights reviewing the data has revealed (section 5).
Specific comments:
(1) Abstract lines 16-18: lists 4 groups of functions for DJ-1, whereas the body of review is organised around 3 groups. It would be better to have the same organisational structure in both.
(2) Introduction: DJ-1 mutations are not mentioned until paragraph 4 (line 60), it would be helpful to introduce the topic of the review (DJ-1) earlier and explain to the reader why understanding this protein could be especially helpful for understanding ALS. Then give the overview of other genetic linkages afterwards to provide context.
(3) Introduction: The introduction is too vague about DJ-1 function. It would be better to set out at high level: what are the genetic and clinical observations linking DJ-1 to ALS, what is the disease model data supporting causal links, and what is the list of known DJ-1 functions. This would set the scene for more detailed discussions in later sections. As written, DJ-1 function is described almost entirely in terms such as: “implicated in multiple cellular functions”, “neuroprotective”, “multimodal”, “moonlighting protein”. The current introduction created an impression that DJ-1 is poorly understood. The information in sections 2, 3, and 4 then seemed surprising that DJ-1 function is better characterised than the introduction led me to expect.
(4) Introduction lines 68 – 69: On first reading, it is difficult to reconcile the observation of increased DJ-1 levels in sporadic ALS patient CSF with the observations that reduced DJ-1 levels contribute to causing disease in the genetic form and in models. It would be helpful to discuss possible interpretations, such as that DJ-1 might be up-regulated as a compensatory response to other drivers of ALS disease and that DJ-1 mutation (or knock-out) might therefore increase sensitivity. In the conclusion section, the authors should discuss possible reasons for why the observed DJ-1 upregulation in sporadic ALS is not sufficient to stop disease progression and mechanisms by which alternative activators of DJ-1 regulated functions might nevertheless be more effective.
(5) Introduction line 62 and lines 90-91: DJ-1 is described both as a ubiquitous protein and also as, highly expressed in the CNS, particularly astrocytes. Although these two descriptions are compatible, they don’t sit easily together. It would be helpful to add some additional discussion of what the higher CNS and astrocytes expression of this ubiquitous protein might reveal about its roles in CNS compared to other tissues and cell types.
(6) Conclusion: In addition to the discussion suggested in point 4 above, it would be helpful to summarise briefly (1) the list of DJ-1 functions for which there is strong evidence, (2) how drugs that up-regulate those functions might potentially help slow or stop ALS disease progression, (3) whether or not there are examples of any such drugs already in development, and (4) if the authors believe such drugs might be able to be effective as single agent therapies or if combination therapies are more likely to be necessary.
Author Response
Reviewer #1
The authors have reviewed the biological functions of the protein DJ-1. The value of this review comes because mutations in the DJ-1 gene are linked with a subset of familial ALS forms. Understanding DJ-1 protein function may therefore help us understand how disrupting those functions contributes to ALS and, more broadly, may offer insight into which disease mechanisms drive all forms of ALS.
The review contains a great deal of valuable information which does indeed offer insights into potential disease mechanisms of ALS. The strongest sections of the review are 2, 3, and 4, which set out evidence supporting DJ-1 involvement in oxidative stress and mitochondrial function, energy metabolism, and response to hypoxia. The greatest room for improvement is in sections 1 and 5. The authors could make these sections more valuable and easier to read by being more concise and precise in setting out why understanding DJ-1 mutation (over other genetic associations) is an especially rich source of insight into general ALS disease mechanisms (section 1); and setting out more clearly which insights reviewing the data has revealed (section 5).
Specific comments:
(1) Abstract lines 16-18: lists 4 groups of functions for DJ-1, whereas the body of review is organised around 3 groups. It would be better to have the same organisational structure in both.
We thank the referee for her/his observation. As mitochondrial dysfunction and oxidative stress are strictly interconnected, for a sake of clarity, in the main text we merged these two topics in the same Chapter. However, following the referee’s suggestion, in the revised version we have modified the abstract as follows: “Here we focus on the involvement of DJ-1 in interconnected cellular functions related to mitochondrial homeostasis, ROS levels, energy metabolism, and hypoxia response, in both physiological and pathological conditions.”
(2) Introduction: DJ-1 mutations are not mentioned until paragraph 4 (line 60), it would be helpful to introduce the topic of the review (DJ-1) earlier and explain to the reader why understanding this protein could be especially helpful for understanding ALS. Then give the overview of other genetic linkages afterwards to provide context.
We understand the reviewer’s point of view, although we would prefer to keep the current organizational structure. In fact, our introduction was intended to present an overview of ALS, mentioning the presence of genetic forms of the disease only to introduce DJ-1, which is the focus of the following chapters. Since the data correlating DJ-1 to ALS mention SOD1 and TDP-43, two largely studied proteins involved in familial forms of ALS, we decided to shortly introduce these proteins before DJ-1.
(3) Introduction: The introduction is too vague about DJ-1 function. It would be better to set out at high level: what are the genetic and clinical observations linking DJ-1 to ALS, what is the disease model data supporting causal links, and what is the list of known DJ-1 functions. This would set the scene for more detailed discussions in later sections. As written, DJ-1 function is described almost entirely in terms such as: “implicated in multiple cellular functions”, “neuroprotective”, “multimodal”, “moonlighting protein”. The current introduction created an impression that DJ-1 is poorly understood. The information in sections 2, 3, and 4 then seemed surprising that DJ-1 function is better characterised than the introduction led me to expect.
We agree with the reviewer that the description of DJ-1 functions in Chapter 1 is a bit vague. This is due in part because too many functions, not always related to ALS, have been ascribed to the protein, and also because we intended to present in the following Chapters the DJ-1 roles which are most relevant in the context of ALS. However, following the reviewer’s suggestion in the revised version we have added a short presentation of the numerous purported roles in Chapter 1 (lines: 98-105).
(4) Introduction lines 68 – 69: On first reading, it is difficult to reconcile the observation of increased DJ-1 levels in sporadic ALS patient CSF with the observations that reduced DJ-1 levels contribute to causing disease in the genetic form and in models. It would be helpful to discuss possible interpretations, such as that DJ-1 might be up-regulated as a compensatory response to other drivers of ALS disease and that DJ-1 mutation (or knock-out) might therefore increase sensitivity. In the conclusion section, the authors should discuss possible reasons for why the observed DJ-1 upregulation in sporadic ALS is not sufficient to stop disease progression and mechanisms by which alternative activators of DJ-1 regulated functions might nevertheless be more effective.
As the reviewer pointed out, the most likely hypothesis to explain the presence of increased DJ-1 levels in CSF of sporadic ALS patients is related to a protective role exerted by the protein in the first stages of the disease. In fact, high levels of DJ-1 have been found in the serum or plasma of patients with breast cancer, melanoma, familial amyloid polyneuropathy, and Parkinson’s disease, and in the CSF of patients with sporadic Parkinson’s disease, and a similar hypothesis has been proposed in the frame of PD (PMID: 20202083). As suggested by the referee, in the revised version of the review a new statement concerning such a hypothesis has been added on lines 72-75. This concept was also taken up in the conclusions.
(5) Introduction line 62 and lines 90-91: DJ-1 is described both as a ubiquitous protein and also as, highly expressed in the CNS, particularly astrocytes. Although these two descriptions are compatible, they don’t sit easily together. It would be helpful to add some additional discussion of what the higher CNS and astrocytes expression of this ubiquitous protein might reveal about its roles in CNS compared to other tissues and cell types.
We understand the reviewer’s observation and, following her/his suggestion, in the revised version of the manuscript we have added a new sentence to explain how the presence of DJ-1 in astrocytes could be related to the disease (lines: 94-98), although such an analysis is out of the aim of this review.
(6) Conclusion: In addition to the discussion suggested in point 4 above, it would be helpful to summarise briefly (1) the list of DJ-1 functions for which there is strong evidence, (2) how drugs that up-regulate those functions might potentially help slow or stop ALS disease progression, (3) whether or not there are examples of any such drugs already in development, and (4) if the authors believe such drugs might be able to be effective as single agent therapies or if combination therapies are more likely to be necessary.
We thank the reviewer for her/his comment and in the revised version conclusions have been largely modified by taking into consideration the referee’s observations.
Reviewer 2 Report
Well written review in idiomatic ENGLISH. The underlying 'message' eventually declares itself as that ALS is associated with impairment of mitochondrial metabolism, or perhaps reserve, taking hypoxic stress as an example. However, there's no evidence for the latter in the crucial early stages of the disease, and work on circulating VEGF changes suggests this is itself secondary to disease-related hypoxia from respiratory muscle involvement. So there's a crucial issue concerning primary and secondary changes. The evidence put forward for DJ-1 dysfunction as a major or even primary factor is weak and highly speculative.
Reworking the review to focus on the evidence for primary (as distinct from hypoxic secondary) changes in mitochondrial function would be more apposite and of greater value - the notion of DJ-1 dysfunction could be worked into such a discussion. It is the onset of ALS that needs to be understood first and foremost, not the late stages.
Line 250 does not read sensibly - there's missing text?
Line 340 "oxidative conditions" is meaningless: Disorded?
Line 359 et seq - this final paragraph is disappointingly vague (see my comments above)
Author Response
Reviewer #2
Well-written review in idiomatic ENGLISH. The underlying 'message' eventually declares itself as that ALS is associated with impairment of mitochondrial metabolism, or perhaps reserve, taking hypoxic stress as an example. However, there's no evidence for the latter in the crucial early stages of the disease, and work on circulating VEGF changes suggests this is itself secondary to disease-related hypoxia from respiratory muscle involvement. So there's a crucial issue concerning primary and secondary changes. The evidence put forward for DJ-1 dysfunction as a major or even primary factor is weak and highly speculative.
Reworking the review to focus on the evidence for primary (as distinct from hypoxic secondary) changes in mitochondrial function would be more apposite and of greater value - the notion of DJ-1 dysfunction could be worked into such a discussion. It is the onset of ALS that needs to be understood first and foremost, not the late stages.
We appreciate the reviewer’s comments, although we think that most of them arise from a misunderstanding. In the original version, we were probably unable to properly communicate our message. In fact, as we stated in the last paragraph of Chapter 1, we fully agree with the reviewer that DJ-1 dysfunction does not represent a major or even a primary factor involved in ALS pathogenesis, and this is not the “take-home message” of the review. The idea is that DJ-1 loss of function may alter some cellular pathways favoring the onset or the progression of the disease only when other primary factors are present. In other words, DJ-1 mutations represent a genetic background that makes individuals more sensitive to other pathological factors. For this reason, the review did not necessarily focus on primary events, but as indicated in the title, we were interested in evaluating DJ-1-related functions that could have somehow relevance in the frame of ALS. Among them, the involvement of DJ-1 in the hypoxic response in our opinion is quite relevant since an impaired capability to respond to hypoxic insults might exacerbate the damage and favor the progression of the disease. Moreover, although we agree with the reviewer that many hypoxic responses in ALS patients derive from respiratory muscle alterations, there are also few data suggesting that VEGF expression levels might represent a contributing factor rather than a consequence of ALS (refs 101-104).
Keeping in mind the reviewer’s criticism, in the revised version we have modified the manuscript as follows:
1) We have slightly modified the last paragraph of Chapter 1 (lines 106-117).
2) the hypoxic pathway has been removed from Figure 1 and a new Figure 2 has been added describing the role of DJ-1 in hypoxia while understating the role of hypoxia as a primary event in ALS onset.
3) In Chapter 4, we have added a new sentence emphasizing that defects in respiratory muscles are mainly responsible for the hypoxic alterations observed in ALS patients (lines 316-318).
Line 250 does not read sensibly - there's missing text?
We have rewritten the sentence (lines: 260-262).
Line 340 "oxidative conditions" is meaningless: Disorded?
We agree with the reviewer that “oxidative conditions” as well as “oxidative stress” are a bit meaningless, but the problem with DJ-1 is that how the protein protects against oxidative damage has not been fully elucidated from a molecular point of view. Anyway, following the referee’s suggestion “oxidative conditions” has been replaced by “redox alterations”.
Line 359 et seq - this final paragraph is disappointingly vague (see my comments above)
In the revised version, the final paragraph has been rewritten (lines: 387-401).
Reviewer 3 Report
In this Review the authors discuss the involvement of DJ-1, a multifunctional protein ubiquitously express, mainly linked to Parkinson’s disease, that functions primarily as an antioxidant through various mechanisms, in the ALS pathogenesis. The rationale resides in several observations that lead to the assumption that DJ-1 although not a primary factor, could contribute to the onset/progression of ALS. They focus on some physiological features linked to DJ-1 function and ALS pathogenesis.
General comments
This is a timely review. It is well-structured and generally well-written. It presents a comprehensive analysis of the possible role played by DJ-1 in motor neuron death. It discusses the different aspects of cellular homeostasis where DJ-1 is involved related to neurodegeneration, in general, and ALS. The quality of the illustration is fair, conclusions are sound.
Specific notes and recommendations
The manuscript will benefit from (1) going into more detail about the two families carrying DJ-1 mutations and their ALS-like symptoms, it will help clarify its putative role in ALS; (2) is DJ-1 considered a marker? Since its levels have been found to be significantly higher in the CSF of ALS patients?; (3) although the illustration is fairly clear and there is growing evidence that ALS has a strong vascular component I would represent ischemia/hypoxia in the figure in a different way, not as a whole brain, but more at the vascular/tripartite synapse.
The final recommendation will be made based on the quality of the revision of the manuscript.
Author Response
Reviewer #3
In this Review the authors discuss the involvement of DJ-1, a multifunctional protein ubiquitously express, mainly linked to Parkinson’s disease, that functions primarily as an antioxidant through various mechanisms, in the ALS pathogenesis. The rationale resides in several observations that lead to the assumption that DJ-1 although not a primary factor, could contribute to the onset/progression of ALS. They focus on some physiological features linked to DJ-1 function and ALS pathogenesis.
General comments
This is a timely review. It is well-structured and generally well-written. It presents a comprehensive analysis of the possible role played by DJ-1 in motor neuron death. It discusses the different aspects of cellular homeostasis where DJ-1 is involved related to neurodegeneration, in general, and ALS. The quality of the illustration is fair, conclusions are sound.
We thank the reviewer for the positive comments!
Specific notes and recommendations
The manuscript will benefit from (1) going into more detail about the two families carrying DJ-1 mutations and their ALS-like symptoms, it will help clarify its putative role in ALS;
In the revised version we have added the required information on lines 60-61 and 66.
(2) is DJ-1 considered a marker? Since its levels have been found to be significantly higher in the CSF of ALS patients?;
High levels of DJ-1 have been found in the serum or plasma of patients with breast cancer, melanoma, familial amyloid polyneuropathy, and Parkinson’s disease (PMID: 20202083). Moreover, a higher amount of DJ-1 was detected in the CSF of patients with Parkinson’s disease (PMID: 20202083). For these reasons, DJ-1 cannot be considered a marker of ALS.
Also following referee #1’s comment #2, in the revised version of the manuscript, we have highlighted that elevated DJ-1 levels might exert a protective role in the first stages of the disease (lines: 72-75).
(3) although the illustration is fairly clear and there is growing evidence that ALS has a strong vascular component, I would represent ischemia/hypoxia in the figure in a different way, not as a whole brain, but more at the vascular/tripartite synapse.
The final recommendation will be made based on the quality of the revision of the manuscript.
We thank the reviewer for the suggestion. However, since in this review we did not discuss the (important) role of glial cells and astrocytes in the disease, it would be a bit misleading to propose a picture, which describes a tripartite synapse. For this reason, taking into consideration the reviewer's suggestion we have removed from Figure 1 the part related to the hypoxic response and we have added a new Figure 2 focused on the role of DJ-1 in the hypoxic response in which the vascular component is mentioned in the VEGF-related pathways.
Round 2
Reviewer 2 Report
I appreciate the writers' sensible consideration of my earlier comments.
I suggest the review is now suitable for publication